# The Transcriptional Profile of *Trichophyton rubrum* Co-Cultured with Human Keratinocytes Shows New Insights about Gene Modulation by Terbinafine

**DOI:** 10.3390/pathogens8040274

**Published:** 2019-11-29

**Authors:** Monise Fazolin Petrucelli, Josie Budag Matsuda, Kamila Peroni, Pablo Rodrigo Sanches, Wilson Araújo Silva, Rene Oliveira Beleboni, Nilce Maria Martinez-Rossi, Mozart Marins, Ana Lúcia Fachin

**Affiliations:** 1Biotechnology Unit, University of Ribeirão Preto-UNAERP, Av. Costábile Romano 2201, Ribeirão Preto 14960-900, SP, Brazil; mofazolin@gmail.com (M.F.P.); josie@unidavi.edu.br (J.B.M.); rbeleboni@unaerp.br (R.O.B.); mmarins@gmb.bio.br (M.M.); 2National Institute of Science and Technology in Stem Cell and Cell Therapy, Center for Cell-Based Therapy, Ribeirão Preto 14051-140, SP, Brazil; kcperoni@gmail.com (K.P.);; 3Department of Genetics, Ribeirão Preto Medical School, University of São Paulo, Ribeirão Preto 14049-900, SP, Brazil; psanches@gmail.com (P.R.S.); nmmrossi@usp.br (N.M.M.-R.); 4Center for Integrative System Biology-CISBi-NAP/USP, University of São Paulo, Ribeirão Preto 14049-900, SP, Brazil; 5Center for Medical Genomics, University Hospital of the Ribeirão Preto Medical School, University of São Paulo, Ribeirão Preto 14015-010, SP, Brazil

**Keywords:** antifungal, allylamine, dermatophyte, ergosterol, ERG1, NGS, resistance

## Abstract

The dermatophyte *Trichophyton rubrum* is the main causative agent of dermatophytoses worldwide. Although a superficial mycosis, its incidence has been increasing especially among diabetic and immunocompromised patients. Terbinafine is commonly used for the treatment of infections caused by dermatophytes. However, cases of resistance of *T*. *rubrum* to this allylamine were reported even with the efficacy of this drug. The present study is the first to evaluate the effect of terbinafine using a co-culture model of *T. rubrum* and human keratinocytes, mimicking a fungus-host interaction, in conjunction with RNA-seq technique. Our data showed the repression of several genes involved in the ergosterol biosynthesis cascade and the induction of genes encoding major facilitator superfamily (MFS)- and ATP-binding cassette superfamily (ABC)-type membrane transporter which may be involved in *T. rubrum* mechanisms of resistance to this drug. We observed that some genes reported in the scientific literature as candidates of new antifungal targets were also modulated. In addition, we found the modulation of several genes that are hypothetical in *T. rubrum* but that possess known orthologs in other dermatophytes. Taken together, the results indicate that terbinafine can act on various targets related to the physiology of *T. rubrum* other than its main target of ergosterol biosynthetic pathway.

## 1. Introduction

Dermatophytoses are superficial infections of keratinized tissues that are caused by a group of filamentous fungi, called dermatophytes [1]. Among these dermatophytes, *Trichophyton rubrum* is the main causative agent of dermatophytoses in the world [2,3]. Despite their restriction to the superficial layers of the epidermis, dermatophytoses have gained particular importance because of the increasing number of cases among diabetic patients [4] and patients using immunosuppressive drugs [5]. Furthermore, knowledge of the molecular mechanisms involved in the fungus-host interaction is still limited because of technical difficulties of the available models that mimic this interaction. However, new models such as co-culture of *T. rubrum* with keratinocyte cell lines and culture media containing keratin and elastin were introduced recently as strategies to evaluate compounds with antifungal activities [6,7,8].

Among commercial compounds, terbinafine, an antifungal agent of the allylamine class, is commonly used for the treatment of infections caused by dermatophytes, with the highest activity being observed against species of the genus *Trichophyton*, *Microsporum*, and *Epidermophyton* [9]. Its main target is the inhibition of the ergosterol biosynthetic pathway through a specific mechanism of action that consists of inhibiting the enzyme squalene epoxidase [10,11]. The inhibition of the early steps of this pathway triggers fungicidal activity against susceptible species, particularly filamentous fungi [11]. However, despite its efficacy, cases of resistance of *T. rubrum* to this allylamine have been frequently reported [12,13,14,15,16]. Studies suggest that the resistance mechanisms of species of the genus *Trichophyton* to terbinafine are associated with specific mutations in the squalene epoxidase gene [14,15] and with membrane transporters that mediate the efflux of the drug [17].

Previous studies showed the effect of terbinafine on the expression of *T. rubrum* genes using microarray [18] and subtractive suppressive cDNA library in culture medium [19]. However, DNA microarrays lack sensitivity for genes that are expressed at lower or very high levels and therefore have a smaller dynamic rang [20]. 

In order to present a more in-depth view of the effect of terbinafine on *T. rubrum* genes modulation, the present study is the first to evaluate the effect of this drug using large-scale analysis of the expression profile of *T. rubrum* co-cultured with human keratinocytes in an attempt to mimic fungus-host interactions. In addition to this model, we applied the dual RNA-seq technique that is widely used for the study of these interactions [21] in bacteria [22], viruses [23], and fungi [24,25,26], and enable the entire transcriptome to be surveyed in a high-throughput and quantitative manner [20]. 

Our data demonstrated the modulation of several *T. rubrum* genes involved in the biosynthesis of ergosterol in addition to the target gene of terbinafine that is already known, genes encoding major facilitator superfamily (MFS)- and ABC-type membrane transporters that are associated with the phenomenon of multidrug resistance, genes that have been discussed in the scientific literature as possible novel targets for antifungal compounds and genes which do not have a known function in *T. rubrum* yet, but have been modulated in response to the drug.

## 2. Results

### 2.1. Viability of HaCaT Keratinocytes after Co-Culture in the Presence of Terbinafine

The lactate dehydrogenase (LDH) assay was carried out to evaluate the viability of keratinocytes after 24 h of co-culture. The percentage of LDH release was 30% for co-culture treated or not with terbinafine, showing that terbinafine does not interfere in cell viability during co-cultivation. Triton X-100 (1%) was used as positive control and resulted in 100% of LDH release. 

### 2.2. RNA-seq Analysis of T. rubrum Genes in Response to Co-Culture in the Presence of Terbinafine

Sequencing generated an average of 34 and 39 million raw reads for the co-culture and co-culture with terbinafine libraries, respectively. Low-quality reads were removed and the remaining reads were aligned to the *T. rubrum* reference genome (CBS 118892) of the Broad Institute’s Dermatophyte Comparative Database and *Homo sapiens* HG19 genome (UCSC Genome Bioinformatics Site, Santa Cruz, CA, USA). An average of 5% and 6.7% quality reads were obtained for the co-culture and co-culture with terbinafine libraries, respectively, when the sequences were compared with the *T. rubrum* genome. The reads generated for all conditions tested, as well as the percentages of alignment of each library, are shown in Appendix A. 

### 2.3. Effect of Terbinafine on the Differential Expression of T. rubrum Genes Co-Cultured with Hacat Keratinocytes

Table 1; Table 2 show the 50 most induced and the 50 most repressed *T. rubrum* genes, respectively, after exposure to terbinafine for 24 h of co-culture. There were 277 *T. rubrum* genes that were differentially expressed after 24 h of co-culture in the presence of terbinafine. Of these genes, 163 were hypothetical or not annotated genes in *T. rubrum*, corresponding to approximately 59% of all modulated genes. Although considered hypothetical in *T. rubrum*, some orthologs were identified in other dermatophytes. It is important to note that some genes discussed in this paper are not listed in Table 1; Table 2 because the fold change values does not reach the 50 most induced or repressed genes. However, the complete list of all induced and repressed genes and their respective orthologs are shown in Appendix A, respectively.

### 2.4. Functional Categorization of Differentially Expressed Genes 

In an attempt to determine which molecular mechanisms were involved during co-culture of *T. rubrum* with HaCaT keratinocytes treated with terbinafine, the differentially expressed genes were categorized according to their molecular functions. The most representative categories are those with *p* < 0.05 and are shown in Figure 1.

The majority of the repressed genes were grouped into functional classes related to ribosomal functions, translation and oxidation-reduction processes. In contrast, most of the induced genes were grouped into functional categories related to ATP-binding, integral to membrane, and transmembrane transport. 

### 2.5. Validation by qPCR 

Fifteen genes were selected for validation: genes involved in ergosterol biosynthesis (TERG_01703; TERG_05717), MFS and ABC transporters (TERG_02822; TERG_08613), genes involved in *T. rubrum* metabolism and nutrient transport (TERG_08278; TERG_02609; TERG_06106; TERG_02542), and some potential antifungal targets that were also modulated in the presence of the drug (TERG_03223; TERG_08480; TERG_03861; TERG_08058; TERG_05698; TERG_02694). 

The gene expression results obtained by RNA-seq showed a strong correlation (r = 0.93, *p* < 0.001) with the gene modulation values obtained by qPCR, demonstrating the reproducibility and accuracy of the sequencing. The comparison of the gene modulation values obtained by qPCR and RNA-seq is shown in Figure 2. 

## 3. Discussion 

Analysis of the transcriptional profile of *T. rubrum* co-cultured with HaCaT keratinocytes in the presence of terbinafine revealed a total of 277 *T. rubrum* genes that were differentially expressed after 24 h of culture. Within this set of genes, approximately 59% are hypothetical or not yet categorized in *T. rubrum.* However, approximately 48% of these hypothetical genes in *T. rubrum* possess orthologs with known functions in other dermatophytes. These findings contribute to further studies in order to identify the function of these hypothetical genes in the terbinafine response, thus enabling the discovery of targets which have not been described for this drug yet. Furthermore, the genomes of seven dermatophytes were sequenced and deposited, including *T. rubrum*, which will permit different types of analysis [27]. However, the assembly of the *T. rubrum* genome is not yet complete and may still be altered [28]. We therefore suggest that the expressive number of hypothetical genes that were found to be modulated in this study might be related to limitations in the annotation of the data currently available for the *T. rubrum* genome. As was observed in this work, a microarray analysis of gene expression in *T. rubrum* cultured in medium with keratin also demonstrated strong induction of genes encoding hypothetical proteins [29].

Based on the sequencing data generated, about 6.7% of the quality reads of the co-culture with terbinafine library could be aligned to the *T. rubrum* reference genome, indicating a predominance of human reads in this library. However, these reads reached coverage of 92.2% of the 8616 annotated *T. rubrum* genes, considering genes with at least one count read. It was expected that a restricted number of reads aligned with the *T. rubrum* genome in the co-culture library, since the difficulty in obtaining equivalent amounts of fungal and human RNA during sequencing of mixed transcriptomes has been discussed in the literature as one of the main challenges since the amount of RNA in the cell differs between the two cell types. Whereas a human cell contains about 20–25 pg RNA, a fungal cell contains only about 0.5–1 pg [21,30].

Regarding the molecular mechanisms involved in the co-culture of *T. rubrum* with keratinocytes in the presence of terbinafine, we observed that the functional categories that grouped the most of the modulated genes were those related to ribosomal functions, translation, oxidation-reduction process, integral component of membrane, membrane transport and ATP-binding. Within the set of genes which were grouped into these functional categories, we chose some of them for validation: serine/threonine (TERG_08278), sucrase ferredoxin (TERG_02609), sulfate permease 2 (TERG_06106), and an integral membrane protein (TERG_02542). In addition to these genes, we validated genes involved in the phenomenon of multidrug resistance, genes that participate in ergosterol biosynthesis, and genes suggested as target candidates for new antifungal drugs, which will be discussed in more detail below.

### 3.1. Transmembrane Transporters and Antifungal Resistance

A major problem in the treatment of fungal infections is the phenomenon of multidrug resistance, which is characterized by the acquisition of chemical and structural resistance to different compounds. Several biochemical mechanisms are involved in the process of multidrug resistance in fungi. The most common mechanisms are reduced absorption of the drug, structural alterations at the target site, and increased drug efflux [31]. The last mechanism is mediated mainly by the action of transmembrane transporters that belong to two superfamilies: the ATP-binding cassette superfamily (ABC) and the major facilitator superfamily (MFS) [32].

In the present study, we observed the induction of seven genes that encode transmembrane transporters belonging to the ABC and MFS superfamily: TERG_00574 (Log_2_ Fold Change: 1.83), TERG_02822 (Log_2_ Fold Change: 1.54), TERG_02912 (Log_2_ Fold Change: 1.89), TERG_05575 (Log_2_ Fold Change: 1.54), TERG_06679 (Log_2_ Fold Change: 1.95), TERG_08613 (Log_2_ Fold Change: 1.63), and TERG_08751 (Log_2_ Fold Change: 2.07). The modulation of these genes might be associated with the efflux of this allylamine and with resistance of *T. rubrum* to terbinafine. 

The efflux of drugs seems to be the main resistance mechanism in dermatophytes [33]. Indeed, Fachin et al. [34] observed an increase in the expression of the TruMDR2 ABC-transporter gene in *T. rubrum* in the presence of terbinafine. Furthermore, Maranhão et al. [35] found that deletion of the TrMDR2 gene in the mutant strain ∆TruMDR2 reduced the infective capacity of *T. rubrum*, characterized by low growth of the fungus on human nails when compared to the growth of the wild-type strain. Taking together to our results, these datas suggests that transmembrane transporters play an important role in the pathogenicity of *T. rubrum*.

### 3.2. The Role of Terbinafine in Inhibition of Ergosterol Biosynthesis

The main target of terbinafine is the enzyme squalene epoxidase, which is encoded by the ERG1 gene, an early step in the late pathway of ergosterol biosynthesis. Thus, antifungal agents of the allylamine class target the ERG1 gene and inhibit the ergosterol biosynthesis, resulting in the accumulation of squalene, which is toxic, and consequently causing the death of the fungal cell [10,33,36].

As expected, we observed in the present study the repression of the ERG1 gene (TERG_05717/ Log_2_ Fold Change: −1.65) (Figure 2). However, our data suggest an important role of terbinafine in the repression of various other genes involved in the ergosterol biosynthesis cascade. In addition to the main target of terbinafine, other genes involved in the biosynthesis of ergosterol were also repressed: C-8 sterol isomerase (ERG2 gene) (TERG_06755/ Log_2_ Fold Change: −1.91), sterol C-24 reductase (ERG4 gene) (TERG_02979/ Log_2_ Fold Change: −2.22), and C-4 sterol methyl oxidase (ERG 25 gene) (TERG_08545/Log_2_ Fold Change: −2.13). Similar results werereported by [18] who evaluated the transcriptional profile of response to terbinafine in *T. rubrum* using a cDNA microarray technique. However, in that case the authors not observed modulation of the ERG1 gene.

We also highlight the repression of TERG_01703 (CYP51) (Log_2_ Fold Change: −2.13) and TERG_04906 (CYP61) (Log_2_ Fold Change: −1.57), which encode lanosterol 14α-demethylase (ERG11) and C-22 sterol desaturase (ERG5), respectively, considered as targets for azolic antifungal agents. The enzyme lanosterol 14α-demethylase is mainly studied in fungi because of its fundamental importance for membrane permeability [37]. In fungi, CYP51 catalyzes the removal of the C-14α-methyl group from lanosterol. This demethylation is considered a checkpoint in the transformation of lanosterol to other sterols, including ergosterol [38]. Since ergosterol is a fundamental component of fungal membranes, many drugs were designed to inhibit the enzyme 14α-demethylase and to prevent the enzyme from functioning [39], such as azole antifungal agents. Significant increases in the transcription of the ERG2, ERG3, ERG5, ERG6, ERG25 and ERG11 genes were described in many fungal species in response to treatment with azolic antifungal agents, including *T. rubrum* [40,41]. 

Considering these results, our data suggest an important role of terbinafine in the repression of various other genes involved in the ergosterol biosynthesis cascade (ERG2, ERG4, ERG5, ERG25, ERG11) besides its main target that is already known.

### 3.3. Genes Identified after Co-Culture of T. rubrum in the Presence of Terbinafine

In addition to the inhibition of genes involved in ergosterol biosynthesis and the induction of transmembrane transporter genes, we found other genes that were modulated during co-culture of *T. rubrum* in the presence of terbinafine that are important for the pathogenicity of this dermatophyte. Some of these genes were identified as possible targets to be explored by new antifungal drugs and will be discussed in more detail below.

#### 3.3.1. Sulfite Efflux Pump SSU1 (TERG_02694)

In dermatophytes, keratinolytic proteases are activated by the cleavage of disulfide bridges through sulfite secretion (sulfitolysis), which is mediated by a sulfite efflux pump. It is important to facilitate the degradation of keratin into oligopeptides and amino acids that will be assimilated by the fungus. In addition, the sulfite efflux pump may also play a role in sulfite detoxification in this dermatophyte [17,42]. Considering its importance during the infection process, we highlight the repression of TERG_02694 (Log_2_ Fold Change: −1.74), which encodes the sulfite efflux pump SSU1 (Figure 2).

The sulfite efflux pump SSU1 is encoded by the *TruSsu1 gene* and was shown by Lechenne et al. [43] that the reduction of disulfide bridges in *T. rubrum* depends on the activity of a sulfite efflux pump encoded by the *TruSsu1* gene, which belongs to the tellurite-resistance/dicarboxylate transporter (TDT) family. Thus, sulfite efflux pumps in keratinolytic fungi may represent new drug targets in dermatology, since they are involved in the activation of keratinolytic proteases that are important for fungus nutrition during infection.

#### 3.3.2. Thioredoxin (TERG_08480)

In addition to the genes encoding sulfite efflux pumps, the thioredoxin gene (TERG_08480; Log_2_ Fold Change: 1.82) has also been proposed as a possible antifungal target [44] and was up-regulated by terbinafine in this work (Figure 2).

In response to fungal attacks, the host initiates a front line defense system, that it includes the production of reactive oxygen species (ROS). It has been observed that some pathogens neutralize the effects of oxidative stress through the synthesis of enzymes and antioxidant molecules such as superoxide dismutases, catalases, peroxidades, thioredoxin, among others [45]. 

Kaufman et al. [46] identified a fungal thioredoxin in *Trichophyton mentagrophytes* using differential cDNA screening technique to identify genes encoding putative virulence factors. The fungal thioredoxin was induced by superficial (keratin) and deep (elastin) skin elements suggesting that the product of this gene could be important in superficial and deep dermatophytosis. In *Cryptococcus neoformans*, the thioredoxin gene is believed to be involved in the mechanism of protection against oxidative stress produced by the host and is considered a virulence factor [44].

There is no evidence yet showing some relation between the thioredoxin gene and terbinafine or allylamine class antifungals. However, we highlight in this paper that thioredoxin gene was up-regulated in the presence of terbinafine when compared to co-culture in the absence of the drug, suggesting that *T. rubrum* probably induces this virulence factor necessary for growth in the host in response to the stress generated by the drug action.

#### 3.3.3. Glycosyl Hydrolase (TERG_02742)

We also highlight the induction of TERG_02742 (Log_2_ Fold Change: 1.72), which encodes a glycosyl hydrolase (Figure 2). Glycosyl hydrolases are classified into more than 45 families [47]. Among these families, chitinases belong to family 18 of glycosyl hydrolases and were chemically and genetically validated in pathogenic fungi as a potential drug target. Chitinases hydrolyze chitin by cleavage of the β-1,4 glycosidic bond and play an important role in the biosynthesis of the hyphal cell wall of filamentous fungi. However, chitinases are also present in the human genome [48].

Further studies still need to be performed to verify which family belongs to glycosyl hydrolase modulated in this work in order to identify its possible function in *T. rubrum* as a new therapeutic target in response to drugs.

#### 3.3.4. Transcription Factor C2H2 (TERG_03861)

The gene that encode the transcription factor C2H2 (TERG_03861; Log_2_ Fold Change: 2.39), also considered as a possible new virulence factor, was induced during co-culture in the presence of terbinafine (Figure 2). The role of this transcription factor in dermatophytes has not yet been elucidated. However, a query of the NCBI (National Center of Biotechnology) database showed that this protein exhibits 100% of similarity with the C2H2 transcription factor described in *Aspergillus fumigatus*. In *A. fumigatus*, the C2H2 transcription factor seems to function as a negative regulator of host cell damage and stimulation, as well as of virulence during invasive pulmonary disease [49]. Moreover, *A. fumigatus* C2H2 showed limited homology with the *Candida albicans* C2H2 Bcr1 transcription factor. In the pathogenic yeast *C. albicans*, the C2H2 Bcr1 transcription factor governs the expression of multiple cell surface proteins that mediate biofilm formation, as well as adherence to host cells [50,51]. The similarity of the C2H2 transcription factors of *A. fumigatus* with the C2H2 transcription factor of *T. rubrum* may suggest another virulence factor that is still unknown in this dermatophyte. However, studies still need to be done in order to verify if the transcription factor C2H2 modulated in this work is a virulence factor for *T. rubrum* and what is its function during the infectious process.

#### 3.3.5. Βeta-lactamase and Metallo-β-lactamase (TERG_ 05698 and TERG_08360)

Other possible virulence factors that could be better studied in fungi are fungal lactamases. In this work, the genes encoding β-lactamase (Log_2_ Fold Change: −2.33) (Figure 2) and metallo-β-lactamase (Log_2_ Fold Change: −1.83) in *T. rubrum* were repressed by exposure to terbinafine 

To combat antibiosis, bacteria developed resistance mechanisms such as efflux pumps and hydrolytic enzymes, including β-lactamases [52]. Parallel to the presence in bacteria, β-lactamase-encoding genes are also abundant in different fungal families. In contrast to bacteria, the function of these genes in fungi is still poorly understood. There are only two studies on the hydrolytic function of lactamase (metallo-β-lactamase, MBL)-encoding genes in *Fusarium verticillioides* and *Fusarium pseudograminearum* [53,54]. These studies raised the hypothesis that, asin bacteria, hydrolytic enzymes in fungi might be related to the degradation of and resistance to xenobiotic compounds [55]. 

Further studies should be carried out to evaluate the role of lactamases in dermatophytes, since the participation of fungal lactamases in dermatophytoses and in response to antifungal agents is still unknown.

#### 3.3.6. N-acetylglucosamine-6-phosphate Deacetylase (TERG_03223)

N-acetylglucosamine (GlcNAc) is the main component of the structural polymers of bacteria, plants, and animals. In eukaryotes, GlcNAc is used as a substrate by chitin synthase, an enzyme that produces chitin, a GlcNAc homopolymer [56,57].

In the present study, we observed the induction of N-acetylglucosamine-6-phosphate deacetylase (TERG_03223; Log_2_ Fold Change: 1.96) (Figure 2). This enzyme catalyzes the deacetylation of GlcNAc. In bacteria, this is an important process for lipopolysaccharide synthesis and cell wall recycling [56]. The pathogenic fungus *Candida albicans* utilizes GlcNAc as a carbon source for growth. Deletion of the CaNAG2 gene, which encodes N-acetylglucosamine-phosphate deacetylase, attenuated *C. albicans* virulence in a mouse model of systemic infection [58]. 

We therefore suggest that the induction of N-acetylglucosamine-6-phosphate deacetylase in the co-culture with terbinafine may indicate a potential virulence factor of *T. rubrum* to maintain the integrity of its cell wall during the infectious process due to effects of this allylamine on the plasma membrane of the fungus.

## 4. Materials and Methods

### 4.1. Trichophyton rubrum, Media and Growth Conditions

The *T. rubrum* strain CBS 118892 (CBS-KNAW Fungal Biodiversity Center, Utrecht, The Netherlands) sequenced by the Broad Institute (Cambridge, MA, USA) was used. The strain was cultured on Sabouraud dextrose agar (Oxoid, Hampshire, UK) for 15 days at 28 ℃. After this period of incubation, the mycelial surface of the fungus was scraped with a sterile spatula to prepare conidia solution in saline (0.9% of NaCl). The conidia solution was filtered on glass wool and the solution concentration was adjusted in a hemocytometer to 1 × 10^7^ conidia/mL.

### 4.2. Keratinocytes, Media and Growth Conditions

The immortal human keratinocyte cell line HaCaT purchased from Cell Lines Service GmbH (Eppelheim, Germany) was used. The cells were cultured on RPMI-1640 medium (Sigma Aldrich, St. Louis, MO, USA) supplemented with 10% fetal bovine serum at 37 ℃ in an atmosphere containing 5% CO_2_. Penicillin (100 U/mL) and streptomycin (100 µg/mL) were added to the culture medium to prevent bacterial contamination.

### 4.3. Chemicals

Terbinafine was purchased from Sigma Aldrich (St. Louis, MO, USA) and a stock solution (1 mg/mL) was prepared by dilution of the compound in ultrapure water.

### 4.4. Co-Culture Assay and Conditions 

HaCaT cells (2.5 × 10^5^ cells/mL) and *T. rubrum* conidia (1 × 10^7^ conidia/mL) were used for the co-culture assay as described by Komoto et al. [6]. To evaluate the effect of terbinafine, 0.0162 µM (¼ of Minimal Inhibitory Concentration value) of this antifungal agent was added to the co-culture as determined by Komoto et al. [6]. As control, keratinocytes were cultured in the presence and absence of terbinafine to evaluate the effect of the antifungal agent on keratinocytes. The assay of each condition was carried out in triplicate in three independent experiments. The viability of HaCaT cells was evaluated after 24 h of co-culture treated or not with terbinafine by quantifying the release of lactate dehydrogenase (LDH) (TOX7 kit, Sigma Aldrich) according to manufacturer instructions and described by Santiago et al. [59]. Triton X-100 (Sigma Aldrich) was used as control. Absorbance was read at 490 nm in a microplate reader (Multiscan FC, Thermo Fisher Scientific, West Columbia, SC, USA).

### 4.5. RNA Isolation and Integrity Analysis

RNA was isolated and its integrity was analyzed as described by Petrucelli et al. [26]. Only RNA with an RNA integrity number (RIN) > 7 was accepted for construction of the cDNA libraries. The selected RNAs were quantified in a Quantus™ Fluorometer (Promega Corporation, Madison, WI, USA) and then used for the library construction. 

### 4.6. Library Construction and Sequencing

The cDNA libraries were constructed in triplicate for each condition: keratinocyte culture, keratinocyte culture with terbinafine, co-culture and co-culture with terbinafine. We compared the library of co-culture with terbinafine vs. the library of co-culture in order to assess the differentially expressed genes between this conditions. We also compared the library of keratinocyte culture vs. the library of keratinocyte culture with terbinafine to verify if the drug had some effect in the modulation of human keratinocytes genes. However, there was no observed gene expression modulation (data not shown). The libraries were constructed using the TrueSeq^®^ RNA Sample Preparation Kit v2 (Illumina, San Diego, CA, USA) according to manufacturer recommendations. The libraries were validated following the Library Quantitative PCR (qPCR) Quantification Guide (Illumina). A pool of 11 pM of each library was applied to each lane of the flowcell (Illumina) and clustering was performed using cBot (Illumina) according to manufacturer instructions.

Single-end and paired-end sequencing was performed on the Genome Analyzer IIx and Hiseq 2000 (Illumina), respectively, according to manufacturer recommendations. The RNA-seq data are deposited in the GEO database [60] under the accession number GSE135095.

### 4.7. Sequence Data Analysis

The sequence data were analyzed as described by Petrucelli et al. [26]. According to the distribution of the genes, those exhibiting statistical significance <0.05 and a log_2_ fold change ≥1.5 or ≤−1.5 were classified as differentially expressed genes (Appendix A). Functional categorization of the *T. rubrum* genes was based on the Gene Ontology (GO) categories [61] using the Blast2GO algorithm. The BayGO algorithm was used for functional enrichment [62] and categories with a *p* value < 0.05 were considered the most representative. 

### 4.8. Validation of the Data

Fifteen genes were validated by qPCR. For reverse transcription, 1 μg of total RNA from the same conditions assayed and used for sequencing was treated with DNAse 1 Amplification Grade^®^ (Sigma Aldrich, St. Louis, MO, USA) to remove any contamination with genomic DNA. Messenger RNA was converted to cDNA using the High-Capacity cDNA Reverse Transcription^®^ Kit (Applied Biosystems, Foster City, CA, USA) according to manufacturer recommendations. The real-time quantitative (RT)-PCR assays were carried out in triplicate using the SYBR Taq Ready Mix Kit (Sigma Aldrich) in an Mx3300 qPCR System (Stratagene, San Diego, CA, USA). The cycling conditions were: initial denaturation at 94 °C for 10 min, followed by 40 cycles at 94 °C for 2 min, at 60 °C for 60 s and at 72 °C for 1 min. A dissociation curve was constructed at the end of each PCR cycle to check the amplification products. The gene expression levels were calculated using the comparative 2−ΔΔCT method. The 18S [63] and β-tubulin [64] genes were used as normalizer genes. The results are reported as the mean ± standard deviation of three experiments. Pearson’s correlation test was used to evaluate the correlation between the qPCR and RNA-seq techniques. The primers used in the qPCR assays are shown in Appendix A.

## 5. Conclusions

The RNA sequencing of the *T. rubrum* co-culture in the presence of terbinafine showed that several other genes of the ergosterol biosynthesis cascade were repressed. We highlight the induction of transmembrane transporters, which may be associated with the efflux of this allylamine. In addition, other genes that could be involved in the pathogenicity of *T. rubrum* were also modulated in the presence of the drug. These findings offer new perspectives for the discovery of novel antifungal targets or even for structural modifications in the terbinafine molecule that would allow to increase its spectrum of antifungal activity. In addition, we observed the modulation of hypothetical genes with still unknown functions in *T. rubrum* but that possess orthologs in other dermatophytes.

However, further studies are necessary to confirm the relationship of these modulated genes with the pathogenicity and resistance of *T. rubrum* to terbinafine.

## Figures and Tables

**Figure 1 pathogens-08-00274-f001:**
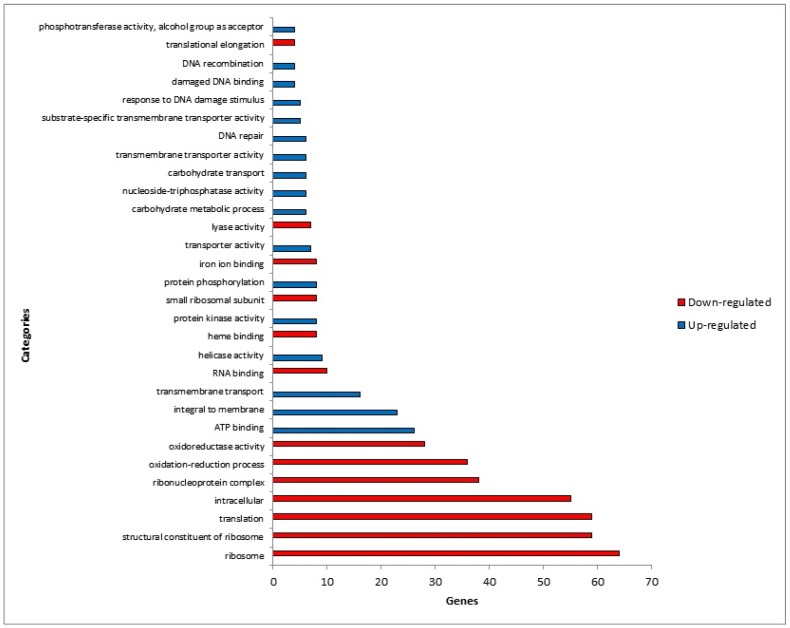
Gene Ontology-based functional categorization of differentially expressed genes. Main representative functional categories (*p* < 0.05) of differentially expressed genes in *T. rubrum* co-cultured with keratinocytes treated with terbinafine.

**Figure 2 pathogens-08-00274-f002:**
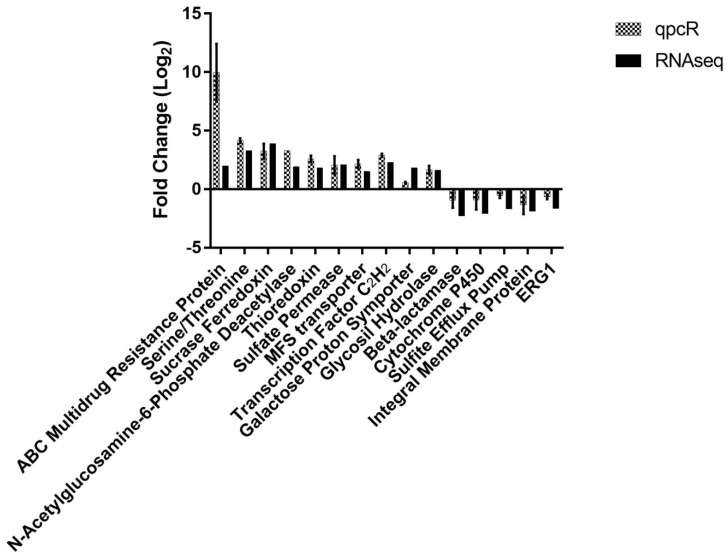
Comparison of gene modulation obtained by RNA-seq and qPCR. The error bars represent the standard error of three independent replicates. Pearson’s test indicated very strong correlation between the two techniques (r = 0.93; *p* < 0.001).

**Table 1 pathogens-08-00274-t001:** The 50 most up-regulated genes in 24 h of co-culture treated with terbinafine.

ID	Log_2_ Fold Change	Gene Product Name	Orthologous
TERG_00523	5.50	Hypothetical protein	-
TERG_08182	4.79	Hypothetical protein	-
TERG_01636	4.72	Hypothetical protein	*Microsporum gypseum* CBS 118893 ADP-ribosylglycohydrolase (1715 nt)
TERG_02902	4.64	Hypothetical protein	-
TERG_00540	4.44	Hypothetical protein	*Trichophyton verrucosum* HKI 0517 oxidoreductase, zinc-binding, putative (1134 nt)
TERG_12029	4.19	Hypothetical protein	-
TERG_02067	4.03	Hypothetical protein	-
TERG_02609	3.93	Sucrase/ferredoxin	*Trichophyton equinum* CBS 127.97 actin patches distal protein 1 (950 nt)
TERG_06347	3.93	Hypothetical protein	-
TERG_11747	3.79	Hypothetical protein	-
TERG_08503	3.45	Hypothetical protein	*Microsporum canis* CBS 113480 secalin (2547 nt)
TERG_08121	3.38	Protein kinase	*Microsporum gypseum* CBS 118893 protein kinase subdomain-containing protein (891 nt)
TERG_08278	3.33	Serine/threonine	*Trichophyton tonsurans* CBS 112818 serine/threonine protein kinase (2270 nt)
TERG_05843	3.32	Hypothetical protein	*Trichophyton equinum* CBS 127.97 F-box domain-containing protein (2405 nt)
TERG_08194	3.24	Hypothetical protein	*Microsporum canis* CBS 113480 serine/threonine protein kinase (2427 nt)
TERG_00197	3.23	Aldose 1-epimerase	*Trichophyton verrucosum* HKI 0517 aldose 1-epimerase family protein, putative (1631 nt)
TERG_02899	3.17	Hypothetical protein	-
TERG_05111	3.12	Hypothetical protein	-
TERG_03293	3.03	Hypothetical protein	-
TERG_03132	2.91	Hypothetical protein	-
TERG_00959	2.79	Hypothetical protein	*Arthroderma benhamiae* CBS 112371 RNA binding protein, putative (3143 nt)
TERG_03252	2.78	Hypothetical protein	-
TERG_03304	2.77	Hypothetical protein	*Trichophyton verrucosum* HKI 0517 AAA family ATPase, putative (2368 nt)
TERG_06402	2.75	Hypothetical protein	*Trichophyton verrucosum* HKI 0517 Ser/Thr protein phosphatase family protein (921 nt)
TERG_02448	2.74	Hypothetical protein	-
TERG_11932	2.74	Hypothetical protein	-
TERG_02303	2.74	Ankyrin repeat protein	*Arthroderma benhamiae* CBS 112371 ankyrin repeat protein (5579 nt)
TERG_04951	2.73	Hypothetical protein	*Trichophyton equinum* CBS 127.97 U-box domain-containing protein (2461 nt)
TERG_06445	2.72	Hypothetical protein	-
TERG_06992	2.69	Pyridine nucleotide-disulfide oxidoreductase	*Trichophyton tonsurans* CBS 112818 pyridine nucleotide-disulphide oxidoreductase (1785 nt)
TERG_02900	2.69	Hypothetical protein	-
TERG_04234	2.66	Hypothetical protein	*Trichophyton verrucosum* HKI 0517 hydrophobin, putative (562 nt)
TERG_00583	2.63	Hypothetical protein	-
TERG_08046	2.56	Hypothetical protein	*Microsporum gypseum* CBS 118893 beta-lactamase (852 nt)
TERG_04721	2.54	Glutamate carboxypeptidase	*Trichophyton equinum* CBS 127.97 glutamate carboxypeptidase (2412 nt)
TERG_05909	2.51	Hypothetical protein	-
TERG_05239	2.46	DNA polymerase lambda	*Trichophyton verrucosum* HKI 0517 DNA polymerase POL4, putative (2133 nt)
TERG_01956	2.46	Hypothetical protein	*Arthroderma benhamiae* CBS 112371 C2H2 finger domain protein, putative (3359 nt)
TERG_06065	2.43	Hypothetical protein	*Trichophyton verrucosum* HKI 0517 conserved glycine-rich protein (903 nt)
TERG_06207	2.39	Hypothetical protein	*Trichophyton verrucosum* HKI 0517 proline oxidase PrnD (1941 nt)
TERG_07034	2.39	Hypothetical protein	-
TERG_03861	2.39	C2H2 transcription factor	*Trichophyton tonsurans* CBS 112818 C2H2 transcription factor (895 nt)
TERG_03443	2.34	Hypothetical protein	*Trichophyton equinum* CBS 127.97 ankyrin repeat protein (1561 nt)
TERG_03628	2.33	Serine/threonine protein kinase	*Trichophyton tonsurans* CBS 112818 serine/threonine protein kinase (2187 nt)
TERG_00642	2.32	Hypothetical protein	*Trichophyton equinum* CBS 127.97 HHE domain-containing protein (552 nt)
TERG_07982	2.32	Hypothetical protein	-
TERG_05469	2.32	Hypothetical protein	-
TERG_12329	2.32	Hypothetical protein	*Trichophyton tonsurans* CBS 112818 Ku70/Ku80 beta-barrel domain-containing protein (2242 nt)
TERG_06990	2.30	Hypothetical protein	-
TERG_02131	2.29	Hypothetical protein	-

**Table 2 pathogens-08-00274-t002:** The 50 most down-regulated genes in 24 h of co-culture treated with terbinafine.

ID	Log_2_ Fold Change	Gene Product Name	Orthologous
TERG_01731	−3.68	Hypothetical protein	-
TERG_11886	−3.49	Hypothetical protein	*Trichophyton tonsurans* CBS 112818 copper radical oxidase (2955 nt)
TERG_06315	−3.32	Hypothetical protein	*Arthroderma benhamiae* CBS 112371 integral membrane protein (1704 nt)
TERG_00499	−3.32	Hypothetical protein	-
TERG_02811	−3.26	Hypothetical protein	*Arthroderma benhamiae* CBS 112371 acetyl xylan esterase (Axe1), putative (955 nt)
TERG_01900	−3.25	Aquaglyceroporin	*Arthroderma benhamiae* CBS 112371 aquaglyceroporin, putative (1058 nt)
TERG_00520	−3.19	Hypothetical protein	-
TERG_03105	−3.16	Hypothetical protein	-
TERG_07234	−2.76	Hypothetical protein	-
TERG_04164	−2.60	Hypothetical protein	-
TERG_07011	−2.58	Hypothetical protein	*Arthroderma benhamiae* CBS 112371 conserved fungal protein (800 nt)
TERG_12474	−2.48	Hypothetical protein	*Trichophyton tonsurans* CBS 112818 ABC-transporter (2471 nt)
TERG_06276	−2.47	Chromate ion transporter	*Trichophyton tonsurans* CBS 112818 chromate ion transporter (1906 nt)
TERG_01901	−2.44	Glycerol kinase	*Trichophyton equinum* CBS 127.97 glycerol kinase (1680 nt)
TERG_00765	−2.38	Hypothetical protein	-
TERG_01406	−2.37	Hypothetical protein	*Trichophyton equinum* CBS 127.97 phospholipase D (939 nt)
TERG_12475	−2.37	Hypothetical protein	*Arthroderma benhamiae* CBS 112371 ABC-transporter, putative (2336 nt)
TERG_07199	−2.35	Hypothetical protein	-
TERG_05698	−2.33	Beta-lactamase	*Trichophyton verrucosum* HKI 0517 transesterase (LovD), putative (1428 nt)
TERG_07539	−2.30	Hypothetical protein	*Trichophyton tonsurans* CBS 112818 multidrug resistance protein (2253 nt)
TERG_11621	−2.27	Hypothetical protein	-
TERG_02161	−2.25	DOC Family	*Microsporum canis* CBS 113480 DOC family protein (456 nt)
TERG_02979	−2.22	Delta(24(24(1)))-sterol reductase	*Trichophyton tonsurans* CBS 112818 Delta(24(24(1)))-sterol reductase (1551 nt)
TERG_07798	−2.22	Hypothetical protein	-
TERG_02722	−2.21	Hypothetical protein	*Trichophyton equinum* CBS 127.97 WSC domain containing protein (2371 nt)
TERG_00754	−2.20	Hypothetical protein	-
TERG_04382	−2.15	C-14 sterol reductase	*Trichophyton tonsurans* CBS 112818 c-14 sterol reductase (1597 nt)
TERG_05808	−2.14	Hypothetical protein	-
TERG_03083	−2.14	3-dehydroquinate synthase	*Trichophyton equinum* CBS 127.97 pentafunctional AROM polypeptide (4908 nt)
TERG_01703	−2.13	Cytochrome P450 51	*Trichophyton equinum* CBS 127.97 cytochrome P450 51 (1790 nt)
TERG_08666	−2.13	Hypothetical protein	-
TERG_08545	−2.13	C-4 methylsterol oxidase	*Trichophyton equinum* CBS 127.97 C-4 methyl sterol oxidase Erg25 (940 nt)
TERG_01676	−2.10	6,7-dimethyl-8-ribityllumazine synthase	*Trichophyton tonsurans* CBS 112818 6,7-dimethyl-8-ribityllumazine synthase (761 nt)
TERG_04041	−2.10	Sad1/UNC domain-containing protein	*Trichophyton verrucosum* HKI 0517 Sad1/UNC domain protein (2625 nt)
TERG_07810	−2.09	Hypothetical protein	*Trichophyton tonsurans* CBS 112818 phospholipase (3533 nt)
TERG_00613	−2.09	Hypothetical protein	-
TERG_04793	−2.04	Cyclin	*Trichophyton tonsurans* CBS 112818 cyclin (1194 nt)
TERG_06265	−2.03	Hypothetical protein	*Trichophyton equinum* CBS 127.97 LPS glycosyltransferase (1368 nt)
TERG_08359	−2.01	FAD-dependent monooxygenase	*Trichophyton tonsurans* CBS 112818 FAD-dependent monooxygenase (1321 nt)
TERG_02842	−2.01	Hypothetical protein	*Trichophyton equinum* CBS 127.97 6-hydroxy-D-nicotine oxidase (1598 nt)
TERG_03204	−2.00	60S ribosomal protein L7	*Trichophyton tonsurans* CBS 112818 60S ribosomal protein L7 (1079 nt)
TERG_07797	−1.98	Isoflavone reductase	*Trichophyton equinum* CBS 127.97 amino acid permease (2474 nt)
TERG_01994	−1.98	Hypothetical protein	*Trichophyton equinum* CBS 127.97 OPT oligopeptide transporter protein (3035 nt)
TERG_00032	−1.98	Mitochondrial dicarboxylate carrier	*Trichophyton tonsurans* CBS 112818 mitochondrial dicarboxylate transporter (1162 nt)
TERG_01604	−1.97	60S ribosomal protein L36	*Trichophyton tonsurans* CBS 112818 60S ribosomal protein L36 (527 nt)
TERG_03148	−1.95	Hypothetical protein	*Trichophyton equinum* CBS 127.97 molybdenum cofactor sulfurase (1551 nt)
TERG_05236	−1.94	60S ribosomal protein L35	*Trichophyton tonsurans* CBS 112818 60S ribosomal protein L35 (1143 nt)
TERG_01399	−1.93	Hypothetical protein	-
TERG_06755	−1.91	C-8 sterol isomerase	*Trichophyton verrucosum* HKI 0517 C-8 sterol isomerase (Erg-1), putative (628 nt)
TERG_02542	−1.91	Integral membrane protein	*Trichophyton verrucosum* HKI 0517 integral membrane protein Pth11-like, putative (1773 nt)

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
