# Peer review of "The Transcriptional Profile of Trichophyton rubrum Co-Cultured with Human Keratinocytes Shows New Insights about Gene Modulation by Terbinafine"

_pathogens, 2019, doi:10.3390/pathogens8040274_

Round 1
Reviewer 1 Report
The manuscript entitled "The transcriptional profile of Trichophyton rubrum co-cultured with human keratinocytes shows new insights about gene modulation by terbinafine" is an interesting contribution to the assessment of molecular mechanisms of response to antifungal substances and may provide insight into the pathways of resistance. The authors clearly present the research hypothesis, show the results and even more than is required discuss the role of activated or inhibited cell pathways in response to terbinafine. The manuscript is well written and easy to read.
I have two main points that can be included in the manuscript: 1. The authors conducted analyzes after 24 hours of co-cultured with terbinafine. In the scientific literature, however, there are reports of varying levels of gene expression in response to an antifungal substance at various times, particularly hours after exposure. Data regarding different levels of gene expression in the same pathway were revealed after 3, 6, 12 and 24 hours of action of an antifungal substance. Did the authors perform tests after 24 hours of exposure to terbinafine after preliminary analyzes that this is the optimal time of exposure?
2. The authors used conidia rinsed from the surface of the T. rubrum colony for co-cultured. Basically, this is in line with the methodology for performing MIC assessment tests in accordance with the CLSI standard. In the pathogenesis of dermatophyte infection, however, one of the stages involves the ingrowth of hyphae in stratum corneum and the role of spores in it is limited. Did the authors not consider performing analyzes with both washed spores and mycelium?
From minor remarks: line 2-3 in Abstract: please check this statement line 67 in Results: I suggest to give accession number or strain info in this place according T. rubrum reference genomeline line 167-173: check the entire paragraph, incorrectly formatted line 274: Candida albicans in Italics line 309: specify the method for obtaining conidiaAuthor Response
Reviewer 1
The manuscript entitled "The transcriptional profile of Trichophyton rubrum co-cultured with human keratinocytes shows new insights about gene modulation by terbinafine" is an interesting contribution to the assessment of molecular mechanisms of response to antifungal substances and may provide insight into the pathways of resistance. The authors clearly present the research hypothesis, show the results and even more than is required discuss the role of activated or inhibited cell pathways in response to terbinafine. The manuscript is well written and easy to read.
I have two main points that can be included in the manuscript:
The authors conducted analyzes after 24 hours of co-cultured with terbinafine. In the scientific literature, however, there are reports of varying levels of gene expression in response to an antifungal substance at various times, particularly hours after exposure. Data regarding different levels of gene expression in the same pathway were revealed after 3, 6, 12 and 24 hours of action of an antifungal substance. Did the authors perform tests after 24 hours of exposure to terbinafine after preliminary analyzes that this is the optimal time of exposure?Authors’ response: Some studies are reported in the scientific literature to evaluate the gene expression response at different times of exposure to antifungal agents. However, most of them evaluate this gene expression based on the growth of the fungus in culture medium, which facilitates the extraction of fungal RNA at different times of exposure. In the present work, the objective was to evaluate the effect of terbinafine on the transcriptional profile of T. rubrum using a co-culture model with human keratinocytes (HaCat cell line), mimicking a superficial infection. Previous experiments of this co-culture model were performed at 3, 6, 12 and 24 hours in our research group, however, as shown in Figure 1 of response letter, no fungal adherence in HaCat cells was observed at times less than 24h.
Figure 1: Co-culture of human keratinocytes (HaCat cell line) with T. rubrum at 3h, 6h, 12h and 24h.
It can be observed in Figure 1 in 24h there is a visible adhesion of the fungus in the keratinocytes, which is an important factor in the modulation of gene expression and to meet the main objective of this study to evaluate the effect of terbinafine in the fungus-host interaction.
Some studies previously published by our research group showed the adherence process of T. rubrum in keratinocytes at 24 hours by optical microscopy (Komoto et al.2015) and by electron microscopy (Petrucelli et al.2018). In this time of co-culture, occurs the penetration of T. rubrum hyphae in the host cell. We emphasize that dermatophytosis is a superficial infection and therefore, the fact that T. rubrum is adhered to keratinocytes (figure 1), moreover the penetration of the hyphae into the keratinocyte showed by electron microscopy (figure 2B) characterizes that time interval of 24h was most adequate to analyze the fungus-host interaction.
Figure 2. The transmission electron microscopy of the Trichophyton rubrum-HaCat co-culture after 24 h. (A) Human keratinocytes (HaCat) keratinocyte as the control (14kx); (B) Co-culture (14kx). The arrow indicates a fragment of T. rubrum hyphae inside the HaCat cells. This figure is published in Petrucelli et al., 2018
In addition, Komoto et al., 2015 e Cantelli et al.2017 also evaluated the effect of compounds with antifungal activity (terbinafine, trans-chalcona, α-solanine, caffeic acid and lycochalcona A) on T. rubrum gene expression when it was co-cultured with human keratinocytes for 24 hours.
Cantelli, B. A. M .; Bitencourt, T.A .; Komoto, T.T .; Beleboni, R. O .; Marins, M .; Fachin, A. L. Caffeic acid and lycochalcone A interferes with the glyoxylate cycle of Trichophyton rubrum. Biomed. Pharmacother. 2017, 0–1, doi: 10.1016 / j.biopha.2017.11.051.
Komoto, TT; Bitencourt, TA; Silva, G .; Beleboni, RO; Marins, M .; Fachin, AL. Gene Expression Response of Trichophyton rubrum during Coculture on Keratinocytes Exposed to Antifungal Agents Evidence-Based Complement Altern. Med. 2015, 2015, 1–7, doi: 10.1155 / 2015/180535
Petrucelli, MF; Peronni Sanches, PR; Komoto, TT; Matsuda, JB; da Silva Junior, WA; Beleboni, RO; Martinez-Rossi, NM; Marins, M.; Fachin, AL Dual RNA-Seq analysis of trichophyton rubrum and HaCat keratinocyte co-culture highlights important genes for fungal-host interaction Genes (Basel) 2018, 9, 362, doi: 10.3390 / genes9070362.
The authors used conidia rinsed from the surface of the T. rubrum colony for co-cultured. Basically, this is in line with the methodology for performing MIC assessment tests in accordance with the CLSI standard. In the pathogenesis of dermatophyte infection, however, one of the stages involves the ingrowth of hyphae in stratum corneum and the role of spores in it is limited. Did the authors not consider performing analyzes with both washed spores and mycelium?Authors’ response: For the co-culture assay, the conidia solution of T. rubrum was first pre-cultured in liquid Sabouraud (SB) medium for 7 hours as described by Komoto et al. 2015. We add this reference (6) in the new version of the manuscript in the Materials and Methods section: HaCaT cells (2.5 × 105 cells / mL) and T. rubrum conidia (1 × 107 conidia / mL) were used for the co-culture assay as described by [6]
[6] Komoto, T. T .; Bitencourt, T.A .; Silva, G .; Beleboni, R. O .; Marins, M .; Fachin, A. L. Gene Expression Response of Trichophyton rubrum during Coculture on Keratinocytes Exposed to Antifungal Agents. Evidence-Based Complement. Altern. Med. 2015, 2015, 1–7, doi: 10.1155 / 2015/180535.
Previous work by our research group showed that after 7 hours of pre-cultivation in SB there is a predominance of T. rubrum conidia with germ tube formation (data not published yet) (please see below in figure 3) . After this period the fungus is centrifuged, washed with saline to remove SB medium, and resuspended in RPMI medium to be inoculated in keratinocyte cells. We emphasize that the same material was used as a control sample (conidia of T. rubrum pre-cultured in liquid SB medium without keratinocytes) for sequencing and gene expression analysis. The pre-cultivation was necessary to obtain larger fungal RNA for the construction of cDNA libraries for sequencing. Other papers published by our group adopted the same co-cultivation methodology with pre-cultivation of T. rubrum:
Cantelli, B. A. M.; Bitencourt, T. A.; Komoto, T. T.; Beleboni, R. O.; Marins, M.; Fachin, A. L. Caffeic acid and licochalcone A interfere with the glyoxylate cycle of Trichophyton rubrum. Biomed. Pharmacother. 2017, 0–1, doi:10.1016/j.biopha.2017.11.051
Petrucelli, M. F.; Peronni, K.; Sanches, P. R.; Komoto, T. T.; Matsuda, J. B.; da Silva Junior, W. A.; Beleboni, R. O.; Martinez-Rossi, N. M.; Marins, M.; Fachin, A. L. Dual RNA-Seq analysis of trichophyton rubrum and HaCat keratinocyte co-culture highlights important genes for fungal-host interaction. Genes (Basel). 2018, 9, 362, doi:10.3390/genes9070362.
Figura 3: Monitoring germination of T.rubrum conidia incubated in Sabouraud medium at 28ºC in 7h. Arrows indicate the growth stage of the conidia. In 7h the beginning of the formation of germ tubes.
From minor remarks:
3) Line 2-3 in Abstract: please check this statement line
Authors’ response: The authors appreciate the suggestion but statement that the incidence of dermatophytosis in diabetic and immunosuppressed patients is in accordance with the references cited in the manuscript:
[4] Gupta, A. K .; Foley, K. A .; Versteeg, S. G. New Antifungal Agents and New Formulations Against Dermatophytes. Mycopathology 2017, 182, doi: 10.1007 / s11046-016-0045-0.
[5] Kaur, R .; Panda, P. S .; Sardana, K .; Khan, S. Mycological Pattern of Dermatomycoses in a Tertiary Care Hospital. J. Trop. Med. 2015, 2015, 1–5, doi: 10.1155 / 2015/157828
In addition, the sentence of lines 2-3 of the abstract: “Terbinafine is one of the most indicated commercial antifungals for the treatment of dermatophytoses” was modified to “Terbinafine is commonly used for the treatment of infections caused by dermatophytes, according to reference [9] Abdel-Rahman, S.; Newland Update on terbinafine with a focus on dermatophytoses. Clin. Cosmet. Investig. Dermatol. 2009, 49, doi:10.2147/ccid.s3690.
4) 67 in Results: I suggest to give accession number or strain info in this place according T. rubrum reference genomeline.
Authors’ response: The authors appreciate the suggestion and the change was made. Please see the new version of the manuscript (Lines 82-83)
5) Line 167-173: check the entire paragraph, incorrectly formatted.
Authors’ response: The authors appreciate the suggestion and the changes have been made. Please see the new version of the manuscript (Lines 190-198)
6) line 274: Candida albicans in Italics
Authors’ response: The authors appreciate the suggestion and the change was made. Please see the new version of the manuscript. (Line 272)
7) line 309: specify the method for obtaining conidia
Authors’ response: Line 309 does not match the edited version. Therefore, this information was included in item 4.1 of Materials and Methods. Please see the new version of the manuscript. (Line 314-317)
Reviewer 2 Report
The present study evaluated the effect of terbinafine using a co-culture model of T. rubrum and human keratinocytes, mimicking a fungus-host interaction, in conjunction with RNA-seq technique. This study offers new perspectives for the discovery of novel antifungal targets or even for structural modifications in the terbinafine molecule that would allow to increase its spectrum of antifungal activity. There are some concerns of this study:
Major concerns:
The authors only validated the results through real-time RT-PCR. They should further overexpress the repressed genes or silence the upregulated gene to see which gene plays an important role in terbinafine antifungal activity. Please revise several sentences that start with a reference number, such as line 175: [34] observed an increase in the expression of the TruMDR2…; line 176: [35] found that deletion of the 176 TrMDR2 genes in the mutant strain…., and many other sentences.Author Response
Reviewer 2
The present study evaluated the effect of terbinafine using a co-culture model of T. rubrum and human keratinocytes, mimicking a fungus-host interaction, in conjunction with RNA-seq technique. This study offers new perspectives for the discovery of novel antifungal targets or even for structural modifications in the terbinafine molecule that would allow to increase its spectrum of antifungal activity. There are some concerns of this study:
Major concerns:
1) The authors only validated the results through real-time RT-PCR. They should further overexpress the repressed genes or silence the upregulated gene to see which gene plays an important role in terbinafine antifungal activity.
Authors’s response: The authors appreciate the reviewer's suggestion and the publication of the gene expression profile may highlight new targets that can be studied from a functionality point of view through deletion or gene overexpression, which may be further developed by our research group in other papers. However, the focus of this work at the moment was to provide a screening of T. rubrum genes that are modulated by terbinafine in a situation that mimics the fungus-host interaction.
Several published papers with transcriptome analysis perform the validation of the data obtained by RT-qPCR:
Bitencourt TA, Macedo C, Franco ME, Assis AF, Komoto TT, Stehling EG, et al. Transcription profile of Trichophyton rubrum conidia grown on keratin reveals the induction of an adhesin-like protein gene with a tandem repeat pattern. BMC Genomics. 2016;17:249. doi:10.1186/s12864-016-2567-8.
Bitencourt TA, Macedo C, Franco ME, Rocha MC, Moreli IS, Cantelli BAM, et al. Trans-chalcone activity against Trichophyton rubrum relies on an interplay between signaling pathways related to cell wall integrity and fatty acid metabolism. BMC Genomics. 2019;20:411. doi:10.1186/s12864-019-5792-0.
Aprianto R, Slager J, Holsappel S, Veening J-W. Time-resolved dual RNA-seq reveals extensive rewiring of lung epithelial and pneumococcal transcriptomes during early infection. Genome Biol. 2016;17:198. doi:10.1186/s13059-016-1054-5.
Meyer FE, Shuey LS, Naidoo S, Mamni T, Berger DK, Myburg AA, et al. Dual RNA-Sequencing of Eucalyptus nitens during Phytophthora cinnamomi Challenge Reveals Pathogen and Host Factors Influencing Compatibility. Front Plant Sci. 2016;7 March:1–15. doi:10.3389/fpls.2016.00191.
Persinoti GF, Peres NT a, Jacob TR, Rossi a, Vencio RZ, Martinez-Rossi NM. RNA-sequencing analysis of Trichophyton rubrum transcriptome in response to sublethal doses of acriflavine. BMC Genomics. 2014;15 Suppl 7:S1. doi:10.1186/1471-2164-15-S7-S1.
We would like to point out that a total of 15 genes were validated and the correlation obtained between the two gene expression analysis techniques was r = 0.93, indicating a strong correlation. This indicates the reliability and reproducibility of the data presented in this work.
2) Please revise several sentences that start with a reference number, such as line 175: [34] observed an increase in the expression of the TruMDR2…; line 176: [35] found that deletion of the 176 TrMDR2 genes in the mutant strain…., and many other sentences.
Authors’ response: The authors agree and the changes were made. Please see the new version of the manuscript. ( Please see the new version of the manuscript. Lines: 178; 179; 229; 242; 332; 336; 340; 360).
Round 2
Reviewer 2 Report
The manuscript has been improved.